# Investigation of Thermophysical Properties of AW-2024-T3 Bare and Clad Aluminum Alloys

**DOI:** 10.3390/ma13153345

**Published:** 2020-07-27

**Authors:** Janusz Zmywaczyk, Judyta Sienkiewicz, Piotr Koniorczyk, Jan Godzimirski, Mateusz Zieliński

**Affiliations:** Faculty of Mechatronics and Aerospace, Military University of Technology, 00-908 Warsaw, Poland; judyta.sienkiewicz@wat.edu.pl (J.S.); piotr.koniorczyk@wat.edu.pl (P.K.); jan.godzimirski@wat.edu.pl (J.G.); mateusz.zielinski@wat.edu.pl (M.Z.)

**Keywords:** 2024-T3 bare and clad alloys, thermophysical properties, viscoelastic properties, activation energy

## Abstract

In this paper, thermophysical and viscoelastic dynamic mechanical measurements (DMA) were performed for bare and clad aluminum AW-2024-T3 alloys. Specific heat, thermal diffusivity, and dynamic module (storage and loss) tests were performed in the range of 50 to 500 °C, except for DMA ones (RT–400 °C). All tests were carried out using the following specialized measuring stands: a light flash apparatus (LFA), differential scanning calorimeter (DSC), and a dynamic mechanical analyzer (DMA). The microstructures and compositions of alloys were investigated by light microscope (LM), scanning electron microscope (SEM), and energy-dispersive X-ray spectroscopy (EDS). Furthermore, Vickers micro-hardness measurements were conducted prior to and after DSC studies. Different precipitation kinetics of the θ′ and S′ metastable phases in the bare 2024-T3 compared to the clad alloy were observed by DSC. Additionally, the DSC results for a few selected scan rates were analyzed by the Kissinger method to give activation energies for the precipitation of θ′ and S′ metastable phases in the alloys. The apparent activation energy of the θ′ and S′ phases corresponds to 137.1 ± 4.4 $\mathrm{kJ}\cdot {\;\mathrm{mol}}^{-1}$ for the bare alloy and 131.0 ± 6.0 (exo) and 104.1 ± 2.1 (exo) (two peaks) for the clad alloy.

## 1. Introduction

Al–Cu based alloys (e.g., Al–Cu–Mg–(Li, Si)) are widely used in aviation technology to build an airframe and other elements of aircraft construction (for example, elements of the wing and fuselage are made of 2024-T3 aluminum alloy sheet). To protect against corrosion, duralumin sheets are often clad with pure aluminum. Non-clad sheets are protected against corrosion by anodizing or allodination. AW-2024-T3 belongs to the well-known 2xxx series of aluminum–copper–magnesium alloys, which are precipitation hardened alloys. The age hardening phenomenon was first discovered in the Al4Cu0.6Mg alloy by Alfred Wilm in 1906 [1]. The 2xxx series of aluminum alloy has a good strength to weight ratio, improved short transverse properties, high resistance against fatigue and corrosion, and reasonably high fracture toughness [2,3]. These good physical and mechanical properties of aluminum alloys result from their microstructures which are influenced by thermo mechanical processing. The effects of aging on the microstructures of the 2024 aluminum alloy have been studied intensively by various authors in the last two decades [4,5,6]. Lin et al. [7] studied the effects of external stress and the creep aging temperature on the hardness and precipitation processes in the 2024-T3 aluminum alloy. Based on the results of experimental research, they came to the conclusion that the precipitation process is very sensitive to external stress and the aging temperature. Prudhomme et al. [8] investigated the effects of aging on the microstructure and physical and mechanical properties of a 2024 aluminum alloy. However, they did not notice significant modification of the fatigue resistance despite changes in the precipitation structure during the service life. Alexopoulos et al. [9] investigated the effect of artificial ageing conditions on the precipitation kinetics and tensile and work hardening mechanical behaviors of aluminum alloy 2024. They found that work hardening stages (under-, peak- and over-aging) are artificially age sensitive. The precipitation sequences in aged 2xxx series alloys are complex, and they are still being investigated by various authors [10]. Gouma et al. [11] focused their attention on the structure of the Guinier–Preston–Bagaryatsky (GPB) zone. Feng et al. [12] used high resolution transmission electron microscopy (HRTEM) and electron energy loss spectrometers (EELS) to show the precipitation sequence of the S-phase during dislocation. However, in the relevant body of literature, there are few works concerning the thermophysical properties of the AW2024 aluminum alloy. An example of this is the work of Mohammadian Semnani and Degischer [13], in which the authors compared the precipitation kinetics of laboratory AlCu4.3 and commercial AlCu4Mg alloys using differential scanning calorimeter (DSC) analysis and the dilatometry technique. 

T3 processing of the AW 2024 alloy consists of supersaturation at 495 °C, followed by cold rolling and natural aging. The clad process involves applying a thin layer of Al 99.8 or 99.7 at 450 °C by rolling on an AW-2024 sheet [14]. The clad process is accompanied by diffusion welding and the occurrence of an intermediate layer. The Al layer thickness is in the order of 25 μm [14]. For AW-2024-T3-cladding, the clad process precedes the T3 heat treatment. The purpose of this work was to compare sequences of precipitation, decomposition, and dissolution in alloys tested by DSC and dynamic mechanical measurement (DMA) methods and to determine the apparent activation energy of the θ′ and S′- metastable phases in the bare 2024-T3 compared to the clad one. Additionally, specific heat, thermal diffusivity, thermal conductivity and dynamic module *E*′, *E*″, tan δ tests were performed. 

## 2. Materials and Methods 

### 2.1. Materials

The as-delivered material was in the form of AW-2024-T3 sheets in both bare as well as clad alloys. The chemical composition (wt.%) of AW-2024 alloy is shown in Table 1. The AW-2024-T3 is a medium to high strength alloy that was solution heat-treated, cold worked, and naturally aged to a substantially stable condition. Al-cladding is essential for ensuring the good tensile and fatigue behavior of the AW-2024 alloy and also plays a role in protecting the AW-2024 alloy against corrosion.

### 2.2. Sample Preparation

Samples for testing thermal diffusivity were in the form of a cylinder with a diameter *d* of 12.70 mm and a thickness *l* of 1.99 mm. These were cut off from a piece of sheet metal by a wire cutting method. In order to ensure high absorption of the pulse generated by the xenon flash lamp, the sample surfaces were covered with a thin layer (2–3 μm) of graphite (GRAPHIT 33, Kontakt Chemie, Iffezheim, Germany). The density of AW-2024-T3 samples measured at room temperature (RT) using a RADWAG analytical balance XA60/220 (Radwag Balances and Scales, Radom, Poland, readability (d): 0.1 mg) was 2.76 g⋅cm^−3^ for bare and 2.79 g⋅cm^−3^ for clad samples, respectively. Samples for DSC investigations had a cylindrical shape with diameter *d* of 3.0 mm and they were encapsulated into a concave Al pan with a volume of 30 μL. The sample mass was 36.04 mg for bare and 50.56 mg for clad AW-2024-T3, respectively. Samples for DMA tests had the shape of a thin strip with dimensions (length × width × thickness) of 55.00 × 8.00 × 1.02 mm^3^, and they were cut off from a piece of sheet by a wire cutting method. Before cross-section observations (SEM), AW-2024 alloy samples were sectioned, mounted in epoxy resin, and metallographically polished. The metallographic samples were etched with Kroll’s reagent. 

### 2.3. Surface Morphology Analysis and Vickers Microhardness Measurements

The morphology of bare and clad aluminum AW-2024-T3 alloys was studied using light microscope (LM), scanning electron microscope (SEM), and energy-dispersive X-ray spectroscopy (EDS). The test samples were of the same sizes as for the DSC measurements, i.e., a diameter *d* of 3.0 mm and thickness *l* of 1.8 mm (bare) or 2.6 mm (clad). Microstructural evaluation was carried out using a KEYENCE VHX-6000 digital microscope (LM, KEYENCE Int., Mechelen, Belgium) and Phenom ProX/CeB6 scanning electron microscope (SEM, ThermoFisher SCIENTIFIC, Eindhoven, The Netherlands) with an acceleration voltage 15 kV equipped with an energy dispersive spectroscopy (EDS) chemical composition analyzer. Grain size was evaluated by image analysis (Image J, National Institutes of Health) using at least 10 SEM cross-sectional images. An X-ray diffraction (XRD) phase analysis was performed using a Rigaku ULTIMA IV diffractometer (Rigaku Americas Corporation., the Woodlands, TX, USA) and Co K_α_ radiation for an angle range (2θ) of 10–130° with steps of 0.02° and a scan speed of 2° per min. The acquired data were processed using the PDF DHN 4 crystallographic database. To validate the mechanical properties of AW-2024 material, Vickers micro-hardness measurements were conducted with a load of 100 g and a loading time of 10 s for each single indentation. The mean value was calculated from ten measurements for every sample.

### 2.4. Thermal Analysis

#### 2.4.1. DSC

The thermal properties were determined using a differential scanning calorimeter DSC 404 F1 Pegasus (NETZSCH-Gerätebau GmbH, Selb, Germany). The temperature range of the DSC investigation was 50–500 °C. As an inert gas, helium was used with a 20 mL⋅min^−1^ flow rate. The specific heat was calculated using the Cp-ratio method based on the 3 DSC curves (baseline, sapphire line, and tested sample line). In order to obtain a stable DSC signal, double evacuation of the helium filling the furnace chamber together with 15-min isothermal segments was applied after each heating/cooling. The heating/cooling rate (HR/CR) was 10 °C⋅min^−1^. The apparent activation energy of the tested samples was determined using the Kissinger free kinetic method [15]. The DSC measurements were carried out at 2, 5, 10 and 20 °C⋅min^−1^ HR. For each fixed heating rate, *β* = d*T*/d*t*, in the DSC measurements, the as-received samples. Based on the Kissinger method, the expression for describing solid-state first-order reactions have the form [15]
(1)$$n\left(\frac{\beta_{m}}{T_{m}^{2}}\right)=ln\left(\frac{AR}{E_{a}}\right)-\frac{E_{a}}{RT_{m}}$$
where *R* = 8.3145 (kJ kmol^−1^ K^−1^) is a universal gas constant, and *A* (K) is the pre-exponential factor of Arrhenius equation.

The apparent activation energy $E_{a}$ (kJ⋅mol^−1^) was calculated on the basis of linear dependence between $ln\left(T_{m}^{2}/\beta_{m}\right)$ and $1000/T_{m}$, where $T_{m}$ (K) stands for the *m*-th temperature peak in the DSC curve. Knowing the directional coefficient of the line, $k_{d}$*,* determined by the linear regression method, the apparent activation energy is given by the formula:(2)$$E_{a}=-8.3145\cdot k_{d},\;\left(\mathrm{kJ}\;\cdot {\mathrm{mol}}^{-1}\right)$$

#### 2.4.2. Light Flash Apparatus (LFA)

The thermal diffusivity was determined using a low-temperature HyperFlash LFA 467 light flash apparatus (NETZSCH-Gerätebau GmbH, Selb, Germany). In this process, the front surface of a plane-parallel sample is heated by a short energy pulse generated by a xenon lamp. From the resulting temperature excursion of the rear face measured with an IR detector, the thermal diffusivity is calculated and, if a reference specimen is also used, the specific heat can also be calculated. The temperature range of the LFA investigation was in the range of 50–500 °C. The following temperature points (in °C) were chosen: for the first heating, 50, 100, 150, 200, 220, 250, 280, 300, 350, 380, 400, and 450, and for the second heating, 40, 120, 180, 230, 260, 290, 310, 330, 360, 390, 420, 430, and 480. As an inert gas, argon with a 20 mL⋅min^−1^ flow rate was used both in heating and cooling modes. At each temperature step during measurement of the thermal diffusivity, three shots were generated to allow an average result to be calculated. As a reference sample for the thermal diffusivity measurement, POCO graphite was used to enable the determination of the specific heat and thermal conductivity of the tested samples by a comparative method. 

#### 2.4.3. DMA

Viscoelastic, i.e., dynamic moduli (storage *E*′ and loss *E*″) tests were performed in the range of 50–400 °C using a DMA 242C dynamic mechanical analyzer (NETZSCH-Gerätebau GmbH, Selb, Germany). The 3-point bending mechanical mode with a frequency *f* of 1 Hz and a HR of 2 °C⋅min^−1^ was used. DMA can be simply described as applying an oscillating force to a sample and analyzing the material’s response to that force [16]. Then, the ability of the material to return or store energy (*E*′), its ability to lose energy (*E*″), and the ratio of these effects (tan δ), which is called damping factor, could be explored. The measurements were performed in an air environment. 

## 3. Results and Discussion 

### 3.1. Al–Cu Phase Diagram

The AW-2024 alloy belongs to the group of precipitation-hardened Al alloys from the 2xxx series (with Cu as a principal alloying element) (Figure 1). 

As can be seen in the Al–Cu phase diagram the solubility limit of Cu in Al is 2.48 ± 0.5 at.% (5.65 wt.%) at a temperature of 548.2 °C (eutectic point on Al-Cu equilibrium diagram), whereas at lower temperatures (below 200 °C), it reduces down to 0.1 at.% [13,18]. For precipitation-strengthened alloys, there must be a terminal solid solution that has a decreasing solid solubility as the temperature decreases. Although the Al–Cu diagram shows one equilibrium θ-Al_2_Cu phase, at least two metastable transition θ″ and θ′ phases precipitate at first [19]. The phenomenon of age-strengthening starts from the natural decomposition of supersaturated solid solution α_sss_ that contains excess Cu and is obtained as a result of homogenizing solution treatment and quenching to room temperature. During the decomposition of α_sss_, Guinier–Preston GP1 and GP2 zones form. GP1 zones, which are composed of Cu-rich aggregation with diameters below 10 nm on the {100} Al plane, develop into GP2 zones, which are 10 nm thick and 150 nm wide plates also with the same lattice structure as the α-Al matrix. The GP2 zone is sometimes called (Al_3_Cu-θ″) [19]. During heating, GP zones dissolve and Cu atoms diffuse to the growing semi-coherent and metastable nuclei of θ″ precipitates and θ′ platelets [13]. At the end, the stable equilibrium θ-Al_2_Cu phase forms. The above-mentioned sequence of precipitation can be written as [13,18,19]: α_sss_ → GP1 zones → GP2 zones → θ″ → θ′ → θ-Al_2_Cu(3)

The authors of papers [20,21] claim that, during natural aging, not all transformations occur, as shown in Formula (3). There are only two stages (4):α_sss_ → GP1 zones → GP2 zones → θ″ → θ-Al_2_Cu(4)

For commercially available AW-2024 alloy, the precipitation sequence differs from the one for the binary system, mostly because of the addition of Mg. According to the work of Semnani et al. [13], this sequence can be written as
α_sss_ → GPB zones → S″ → S′ → S-Al_2_CuMg(5)
with GPB being the Guinier–Preston–Bagaryatsky zones (Cu–Mg co-clusters), S″ and S′ being the metastable phases of Al_2_CuMg, and S-Al_2_CuMg being equilibrium (stable) precipitates. It should be noted that different phases with different distributions may be obtained depending on the chemical composition of AW-2024 especially the Cu content and Cu/Mg ratio. 

### 3.2. As-Delivered State of Bare and Clad AW-2024 

The microstructures of as-received bare and clad AW-2024-T3 alloys are shown in Figure 2a–d. In micrographs obtained by a digital microscope, the multi-phase microstructure composed of α-Al matrix with a range of precipitations dispersed over the microstructure of the Al alloy was made visible (see Figure 2a,b). The α phase corresponds to the solid solution of Cu and the rest of the additives in the face centered cubic (FCC) lattice of Al. The observed precipitations indicate the Al-Cu-X intermetallic compounds [22,23,24,25,26,27]. The microstructure of bare AW-2024-T3 consists of equiaxed grains of the α phase with a size of around 50 µm (Table 2, Figure 2a,b), whereas clad AW-2024-T3 grains exhibit elongation with a large variation around the average values (Figure 2c,d). It was also revealed that the elongation ratio defined as the length/width is higher than 2 (Table 2). Elongated grains are the result of the rolling of the plate over cladding. Typical coarse intermetallic phases for both bare and clad AW-2024-T3, detected by backscattered electron mode with SEM and quantified by EDS, are also shown in Figure 2b,d. In the EDS (Figure 3 and Figure 4, Table 3) analysis, the intermetallic precipitations observed for both AW-2024-T3 bare and clad alloys are various particles with high concentrations of Al, Cu, Mn, and Fe, and they are also enriched by Si.

It should be mentioned that among the coarse intermetallic phases, the equilibrium precipitates and the insoluble constituent ones may be distinguished. Normally, the first precipitates that form during solidification of Al ingots are constituent ones that are enriched in Fe and/or Si impurities [1,28] with the size depending on the solidification rate and the manufacturing process. During further heat treatment, such particles remain insoluble and they are not responsible for strengthening in the precipitation hardened Al alloys. For the studied AW-2024-T3, both bare and clad alloys have irregularly shaped particles, denoted as 1 in Figure 2b and 1 in Figure 2d, and contain various proportions of Al, Cu, Mn, Fe, and Si corresponding to α-Al(Fe, Mn)Si or α-AlMnSi constituent phases [28]. Further, equilibrium precipitates that appeared to be θ-Al_2_Cu and S-Al_2_CuMg and arose during aging after heat treatment of the solution and quenching were detected as round and bright particles (denoted as 3, 4 in Figure 2b and 2 in Figure 2d) [28,29,30,31]. Moreover, black precipitates rich in Al, Mg, and Si (denoted as 2 in Figure 2b) were found in bare AW-2024-T3 alloy. Additionally, the presence of submicronic ovoid particles homogenously distributed throughout the Al matrix were found. Such dispersoids are supposed to be composed mainly of Cu. These precipitates participate in the hardening of alloys, and their role is to hinder dislocation [1,8,9,10,11,12,13,14,15,16,17,18,19,20,21,22,23,24,25,26,27,28,29,30]. It is worth noting that a larger number of intermetallic particles that were also bigger in size and arranged in accordance with the rolling direction were observed for clad AW-2024-T3 alloy. Notwithstanding, a far greater number of dispersoids were detected in bare AW-2024-T3 alloy.

The presence of the precipitates such as MgCuAl_2_ and Al_2_Cu in the bare AW-2024-T3 alloy in the initial state was confirmed through X-ray diffraction (XRD) patterns (see Figure 5). The peaks originating from the α-Al phase, of which the matrix is composed, were also detected on the XRD patterns.

### 3.3. Bare and Clad AW-2024 Microstructures after DSC Heating

The microstructures of as-heated bare and clad AW-2024-T3 alloys (after DSC measurements) are shown in Figure 6a–d. As can be seen in Figure 6, changes in the microstructure of both states of the AW-2024-T3 alloy took place during exposure to elevated temperatures during DSC measurements which indicates that some reactions occurred. Heating during DSC measurement resulted in more precipitation throughout the Al matrix for bare and clad AW-2024-T3 alloys. Bare AW-2024-T3 alloy with irregularly shaped particles denoted as *5, 6,* and *8* in Figure 6b possess high contents of Al and Cu with a small addition of other alloying elements, while particles denoted as 7 in Figure 6b are rich in Si. Clad AW-2024-T3 alloy precipitate denoted as 5 in Figure 6d contain high amount of Cu, whereas precipitate denote as 7 contain a lot of Si. Both precipitates (5 and 7 in Figure 6d) exhibit a low amount of Al. Precipitates denoted as 6 and 8 in Figure 6d are composed mostly of Al with the addition of the rest of the elements. It should be noted that the high concentration of Al may be because these precipitates are quite small and signals collected during EDS analysis may come from the matrix. For bare AW-2024 alloy, dispersoids were also found to be in much greater concentrations in the alloy after DSC testing. This, in turn, means that there are higher exothermic reaction peaks after DSC measurement. Nevertheless, the endothermic (dissolution) and exothermic (precipitation) reactions were shown to be incomplete solid solutions, which demonstrates the occurrence of partial solid solutions. It is worth noting that microstructures of both bare and clad AW-2024-T3 alloys exhibit clear visible features that are characteristic of over-aging. Looking at the microstructures in both LM and SEM images, the post-DCS measurement conditions presented coarser precipitation at the grain boundaries compared with the initial state. Furthermore, precipitates inside grains were even coarser, indicating that over-aging had taken place. According to the EDS analysis (Table 3), newly grown precipitates were enriched mostly in Fe but also in Cu, Mg Mn, and Si. Furthermore, partial recrystallization occurred during DSC measurement for the clad AW-2020-T3 alloy.

Results of the Vickers hardness/microhardness tests are presented in Table 2. It can clearly be seen that after DCS testing, the hardness of analyzed Al alloys notably decreased from 118 HV and 125 HV in the initial state to 70 HV and 59 HV for the bare and clad states, respectively. The decrease in hardness was caused by changes that occurred during heating in DSC testing. The softening of the analyzed Al-alloy appeared as a result of dissolution of the hardening GPB zones as well as some of undissolved Al_2_Cu and Al_2_CuMg intermetallic phases into solid solution [32].

### 3.4. Investigation of Thermal Properties

Thermophysical properties, i.e., the specific heat and thermal diffusivity of bare and clad AW-2024-T3 alloys were studied in the temperature range of about 50–500 °C at least twice. The first heating run allowed the kinetics of the precipitation processes, decomposition sequences, and dissolution precipitation to be identified. The second and subsequent runs of DSC and LFA in the same temperature range did not differ much from each other. This means that at the end of the first run, i.e., at a temperature of about 500 °C, the AW-2024 alloy structures obtained in the T3 and plating processes were removed and both bare and clad materials were obtained in the same heat treatment state, i.e., partial solid solution. The thermophysical properties determined from the second and subsequent runs, i.e., the specific heat and thermal diffusivity of both alloys, bare and clad AW-2024-T3, as a function of temperature are practically the same. It also means that the aluminum layer with a thickness of about 25 µm applied in the plating process on both sides of the plate is thin enough that it does not affect the temperature dependence of specific heat and thermal diffusivity of the clad alloy AW-2024-T3. 

#### 3.4.1. DSC Investigation

DSC thermograms for the first and second heating runs for the AW-2024-T3 and clad AW-2024-T3 alloys are shown in Figure 7 and Figure 8. The second run did not identify exothermic or endothermic heat transitions. The nature of these heating runs is typical for isotropic solids. The combination of the first and second passes allowed the identification of the temperature ranges of the first run in which dissolution and precipitation processes occur. So, for the AW-2024-T3 alloy (Figure 7) in the temperature range of 150–245 °C, in the endothermic process, the dissolution of GP and GPB zones occurred. In the temperature range from 245 to 285 °C, there was θ′ and S′ phase precipitation. Above 285 °C, dissolution of phase precipitations occurred. At the end, a stable equilibrium of the θ-Al_2_Cu and S-Al_2_CuMg phases formed. A small peak at around 390 °C at HR = 10 °C⋅min^−1^ is visible in Figure 7, but was not confirmed by any other DSC signals obtained with a different HR (Figure 9).

A similar characteristic of the DSC thermogram was observed for the AW-2024-T3 clad alloy (Figure 8). In the temperature range of 100 to 225 °C, the GP and GPB zones dissolved. In the temperature range from 225 to 305 °C, precipitation of the θ′ and S′ phases took place. We observed two exothermic peaks at 245.6 and 280.2 °C separated by a local peak at 262.3 °C which was not an endothermal peak but a relative maximum between two exothermal peaks. Thus, we can talk about merging precipitation peaks. Above 305 °C the precipitations of the θ′ and S′ phases dissolved. Finally, for the AW-2024-T3 alloy, a stable equilibrium θ-Al_2_Cu and S-Al_2_CuMg phases formed. Additionally, the DSC results from scan rates were analyzed by the Kissinger method to give activation energies for the precipitation of θ′ and S′ metastable phases in both alloys [33]. For each HR, a new sample was taken for testing. The apparent activation energy of the θ′ phase was 137.1 ± 4.4 kJ⋅mol^−1^ for the bare alloy and 131.0 ± 6.0 (exo) and 104.1 ± 2.1 (exo), (two peaks, Figure 10) for the clad alloy. 

For both alloys, i.e., bare and clad AW-2024-T3, a similar activation energy was obtained, i.e., 137.1 kJ⋅mol^−1^ for bare and 131.0 kJ⋅mol^−1^ for clad. In the case of the AW-2024-T3 clad alloy, there was an overlapping in the precipitation formation process for θ′ and S′. Therefore, for the AW-2024-T3 clad alloy, the activation energy was additionally calculated using the other two peaks, i.e., at 277.8 °C and 302.1 °C. These values should be treated as approximate ones, because they are associated with the overlapping of precipitation formation processes. The enthalpy determined from the DSC scan during the first heating corresponding to the main exothermic peak was equal to 22.7 J⋅g^−1^ for bare AW-2024-T3 and 19.6 J⋅g^−1^ for clad AW-2024-T3. A sensitivity calibration curve was obtained using a few reference materials (In, Sn, Bi, Zn).

#### 3.4.2. LFA Investigation

Temperature tests of thermal diffusivity were carried out during both heating and cooling of the measuring sample. The temperature characteristics of thermal diffusivity for bare and clad AW-2024-T3 alloys obtained from the first heating runs are shown in Figure 11. In the temperature range from about 270 °C, both characteristics were virtually the same. In the temperature range of up to about 300 °C, thermal diffusivity increased as a function of temperature. In the range of 300–400 °C, changes in thermal diffusivity were small, and then a decrease of about 6% in relation to the maximum value was observed.

Figure 12 shows the temperature characteristics of thermal diffusivity for bare and clad AW-2024-T3 alloys obtained from the first cooling run and the second heating run. After the first heating, the material was obtained in a state of so-called partial solid solution. The nature of the dependence of thermal diffusivity as a function of temperature changed. It was the same as for pure aluminum (cf, [34,35,36] for pure Al), i.e., thermal diffusivity decreased linearly as a function of temperature (cf. [37] Figure 3.2.3.0. Effect of temperature on the physical properties of 2024 aluminum alloy for AW-2024-T3 bare). Subsequent heating and cooling processes did not differ from each other and also led to a linear relationship of thermal diffusivity as a function of temperature; this was the same for both alloys. Measurements of thermal diffusivity of bare and clad AW-2024-T3 alloys were performed simultaneously with measurements of thermal diffusivity of the reference material (POCO graphite), which allowed the temperature characteristics of specific heat and thermal conductivity for both tested alloys to be determined by the comparative method [38]. The results are illustrated in Figure 13, Figure 14 and Figure 15 (cf. [37] Figure 3.2.3.0 for bare AW-2024-T3 and [36] for thermal conductivity of pure Al). The maximum relative discrepancies did not exceed about 10% with respect to data in the literature [36,37].

The specific heat and thermal conductivity as a function of temperature for bare and clad AW-2024-T3 alloys had a fixed density at RT, i.e., 2.76 g⋅cm^−3^ for bare alloy and 2.79 g⋅cm^−3^ for clad alloy, as was determined by the comparative method. The temperature characteristics of the apparent specific heat of both alloys obtained by the DSC method and the LFA comparative method for the second heating were similar. The differences in the apparent specific heat $c_{p}$ shown in Figure 13 and Figure 14 were due to the relatively low sensitivity of the LFA comparative method. The thermal conductivity *k* shown in Figure 15 was calculated using the LFA comparative method as a product of the density, thermal diffusivity, and apparent specific heat to obtain data necessary for calculating heat transfer problems.

### 3.5. DMA Investigation

The results of the loss modulus *E*″ test as a function of the temperature for the first and second heating runs are presented in Figure 16. The tests were carried out in the temperature range from RT to 400 °C. Similar to DSC, a qualitative difference between the first and second runs was found. In the temperature range from about 150 to 250 °C, for the first heating run for both tested samples, we observed an increase in the value of the loss modulus *E*″ associated with the dissolution of the *GP* and GPB zones. In the temperature range from 250 to 29 °C for the AW-2024-T3 bare alloy and from 250 to 280 °C for the AW-2024-T3 clad alloy, the precipitation phases of θ′ and S′ occurred. The process was similar for both alloys and the precipitation process only reached 290 °C (inversely in the case of DSC tests) for the AW-2024-T3 bare alloy, and for the AW-2024-T3 alloy, the clad was 10 °C lower, i.e., 280 °C. Then, along with the increase in temperature, dissolution of the precipitations of the θ′ and S′ phases occurred. The process reached 305 °C for the AW-2024-T3 clad alloy and 310 °C for the AW-2024-T3 bare alloy. Above these temperatures, there was a sharp decrease in the loss modulus *E*″ (Figure 16), which was associated with a decrease in the material stiffness of both alloys. The behavior of the loss modulus *E*″ indicates that the process of precipitation of the θ and S fine phases (of the order of 10 nm) occurs, leading to the material of both AW-2024 alloys achieving the state of a so-called partial solid solution. Above 350 °C, a sharp decrease in the storage modulus *E*′ was observed (Figure 17). For the second and subsequent heating runs, the loss modulus *E*″ did not change much and reached an average value of about 3 GPa. Similar behavior to that of the loss modulus *E*″ was observed for tan δ defined as the ratio of *E*″/*E*′, which is presented in Figure 17. It is associated with the monotonic decrease in the storage modulus *E*′ in the range of up to 350 °C. 

## 4. Conclusions

Clad AW2024 sheets are used to cover airframes (pure aluminum plating is an anti-corrosion protector). Bare AW2024 sheets are used for the production of airframe skeletal structures (ribs, frames, stringers, girder elements). These elements are protected against corrosion by anodizing. Both types of sheet are manufactured for the aviation industry and are commercially available. 

In this study, two AW-2024 alloys, one treated with T3 and the other (clad) treated then with T3, were examined. In thermophysical and viscoelastic measurements, the effect of the thin aluminum layer could be neglected due to the very small thickness of 25 µm. All thermophysical and viscoelastic tests were performed for at least two heating runs. The first heating run “took off” the history of heat treatment of both alloys. The temperature characteristics of the thermophysical and viscoelastic properties obtained during the second heating were similar to each other. 

The microstructure analysis confirmed the presence of intermetallic precipitates corresponding mainly to Al_2_Cu and Al_2_CuMg in both alloys in the initial states. These precipitates are responsible for the strengthening mechanism of the alloys. The observed precipitations were also enriched in Mn, Fe, and Si. Noticeable microstructural changes related to the over-aging of both alloys appeared after DSC measurements. For the clad Al-2024-T3 alloy, in addition to the formation of coarser precipitates both on grain boundaries and inside grains, Al-matrix grains transformed from elongated (L/W ratio of 2) to equiaxed ones. For bare AW-2024-T3 alloy, the hardness tests showed an approximately 40% decrease in microhardness after DSC testing that was caused by the occurrence of coarser precipitations that are less likely to strengthen the material than finer precipitates. For clad AW-2024-T3, the decrease in hardness was even greater (over 50%) and was mainly associated with the recrystallization of introduced deformation during Al-plating. 

DSC studies revealed differences in the kinetics of precipitation processes, decomposition, and dissolution of metastable phases in bare and clad AW-2024-T3 alloys. In the case of the AW-2024-T3 clad alloy, there was an overlap in the process of forming θ′ and S′ precipitations (overlapping formation of θ′ and S′). For both alloys, i.e., bare and clad AW-2024-T3, a similar activation energy was obtained, i.e., 137.1 kJ⋅mol^−1^ for the bare alloy and 131.0 kJ⋅mol^−1^ for the clad alloy [13,39].

LFA research revealed slight differences in the thermometric characteristics of the thermal diffusivity of both alloys in the range up to 270 °C. In the temperature range above 270 °C, both had similar characteristics. It should be noted that the thermometric characteristics of the specific heat and thermal conductivity were determined for the second heating run.

DMA studies confirmed the high sensitivity of the method for detecting the kinetics of thermal effects. The viscoelastic properties of both alloys proved to be similar. In the temperature range above 300 °C, DMA tests revealed the further processes of dissolution and precipitation, which were not observed on DSC thermograms.

The results of the investigations of the microstructures and compositions of bare and clad aluminum AW-2024-T3 alloys are summarized as follows:In the initial state, Al_2_Cu and Al_2_CuMg intermetallics enriched in Mn, Fe, and Si were confirmed in both alloys;After DSC measurements, precipitates were coarser and were present inside grains as well as on grain boundaries;A significant decrease in hardness was observed for both alloys after DSC tests.

The results of the kinetic investigations of both alloys recorded during heating confirmed the precipitation/dissolution sequences within the following temperature ranges:
DSC thermograms:
(a)At 150–245 °C (bare) and 100–225 °C (clad), GP and GPB zones dissolve;(b)At 245–285 °C (bare) and 225–305 °C (clad), the θ′ and S′ phases precipitate;(c)At 285–390 °C (bare) and 305–340 °C (clad), the θ′ and S′ phases dissolve;(d)Above 390 °C (bare) and 340 °C (clad), stable equilibrium θ-Al_2_Cu and S-Al_2_CuMg phases form.
DMA thermograms:
(a)At 150–250 °C (bare and clad), GP and GPB zones dissolve;(b)At 250–290 °C (bare) and 250–280 °C (clad), the θ′ and S′ phases precipitate;(c)At 290–310 °C (bare) and 290–305 °C (clad), the θ′ and S′ phases dissolve;(d)At 310–350 °C (bare) and 305–350 °C (clad), the fine phases (~10 nm) of θ and S precipitate;(e)Above 350 °C (bare and clad), stable equilibrium θ-Al_2_Cu and S-Al_2_CuMg phases form and a rapid decrease in the stiffness of the bare and clad alloys (*E*′) is observed.


The results of the thermal property studies are summarized as follows:
LFA thermograms for the first heating:
(a)At 50–300 °C (bare and clad), there is a monotonic increase in thermal diffusivity from 53 mm^2^⋅s^−1^ (bare) and 57 mm^2^⋅s^−1^ (clad) up to 64 mm^2^⋅s^−1^ (for both alloys);(b)At 300–450 °C (for both alloys), there is a monotonic decrease in thermal diffusivity down to 60 mm^2^⋅s^−1^;
LFA thermograms for the second and subsequent heating:
(a)At 100–430 °C (for both alloys), there is a monotonic decrease in thermal diffusivity from 74 mm^2^⋅s^−1^ down to 58 mm^2^⋅s^−1^;
Specific heat determined from LFA investigations (second and subsequent heating) using the comparative method and from DSC studies:An increase in the specific heat value was observed within the temperature range of 100–480 °C, and this was comparable for both alloys. Slight discrepancies between DSC and LFA specific heat results that did not exceed 10% were observed; Thermal conductivity from LFA investigations (second and subsequent heating) using the comparative method:In the temperature range of 100–480 °C (for both alloys), the thermal conductivity changed slightly from 190 to 200 W⋅m^−1^⋅K^−1^. 

## Figures and Tables

**Figure 1 materials-13-03345-f001:**
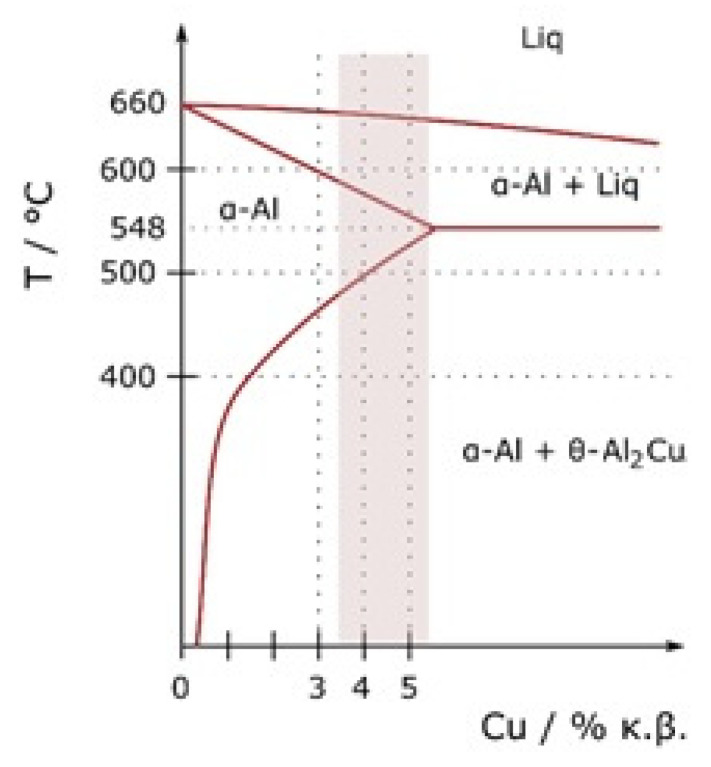
Al–Cu phase diagram [17].

**Figure 2 materials-13-03345-f002:**
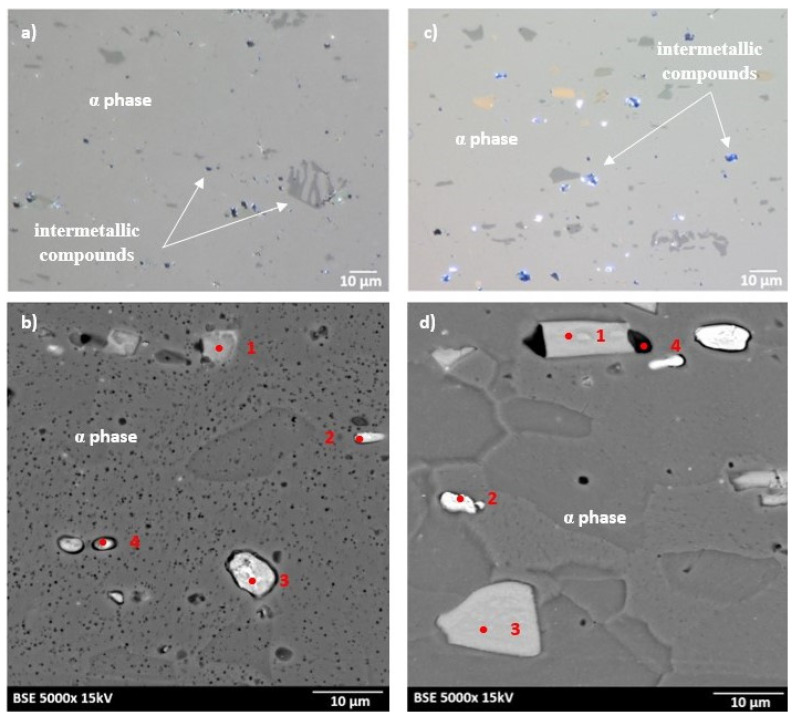
Microstructure of the AW-2024-T3 alloy in the initial state: (**a**) microstructure of bare AW-2024-T3 obtained by light microscope (LM); (**b**) microstructure of bare AW-2024-T3 obtained by SEM; (**c**) microstructure of clad AW-2024-T3 obtained by LM; (**d**) microstructure of clad AW-2024-T3 obtained by SEM.

**Figure 3 materials-13-03345-f003:**
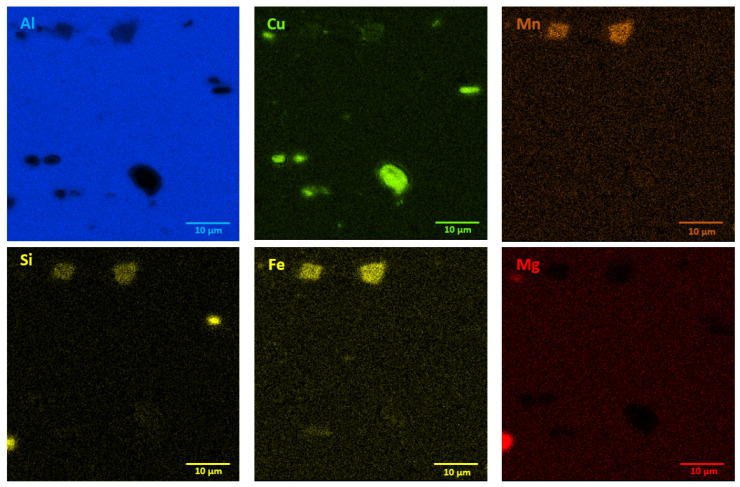
EDS mapping showing precipitates in bare AW-2024-T3 alloy in the initial state.

**Figure 4 materials-13-03345-f004:**
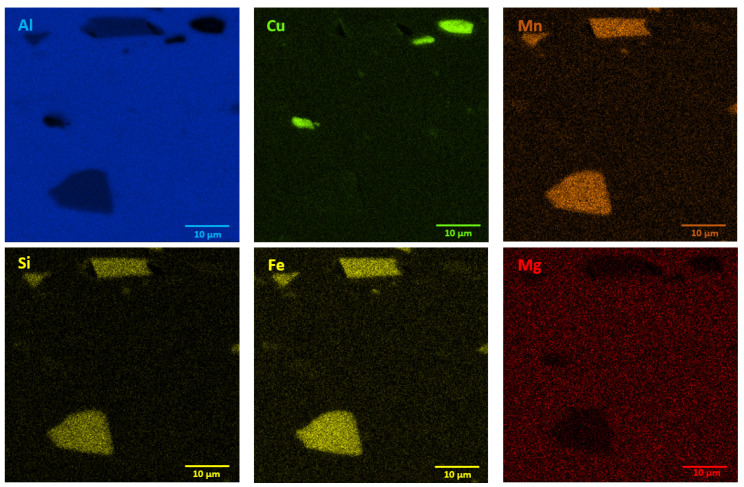
EDS mapping showing precipitates in clad AW-2024-T3 alloy in the initial state.

**Figure 5 materials-13-03345-f005:**
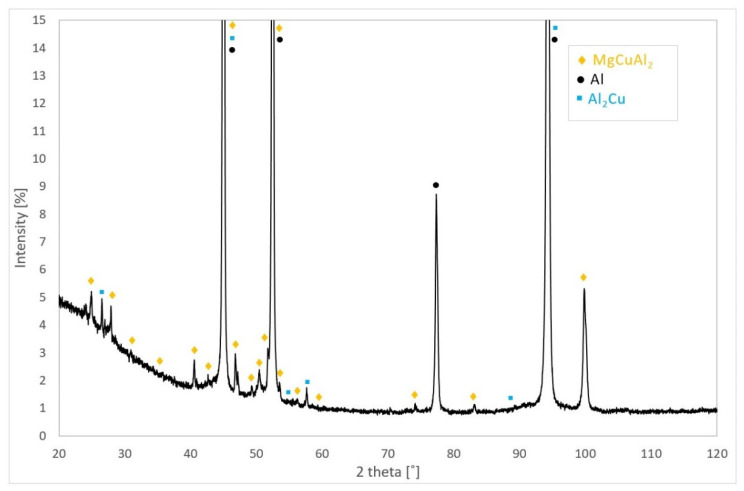
X-ray diffraction (XRD) phase analysis results for bare AW-2024-T3 in the initial state.

**Figure 6 materials-13-03345-f006:**
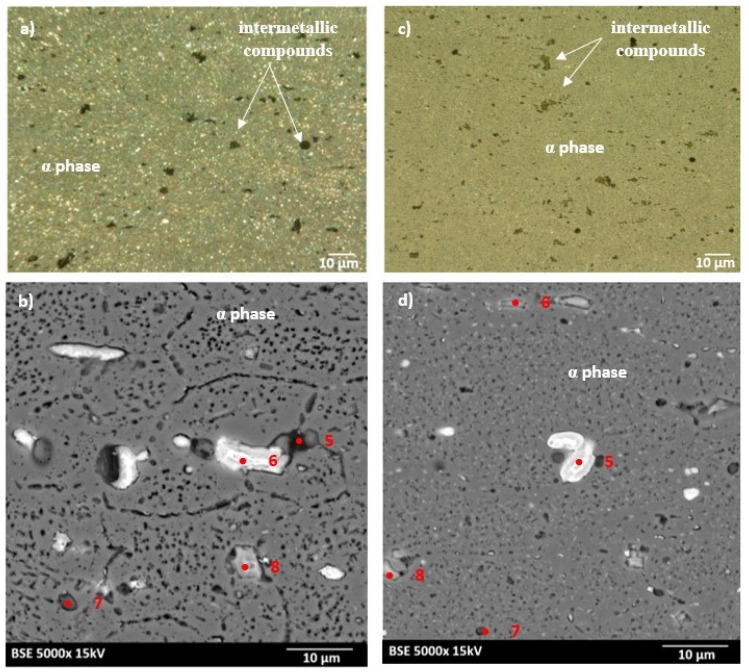
Microstructure of the AW-2024-T3 alloy after DCS measurements: (**a**) microstructure of bare AW-2024-T3 obtained by LM; (**b**) microstructure of bare AW-2024-T3 obtained by SEM; (**c**) microstructure of clad AW-2024-T3 obtained by LM; (**d**) microstructure of clad AW-2024-T3 obtained by SEM.

**Figure 7 materials-13-03345-f007:**
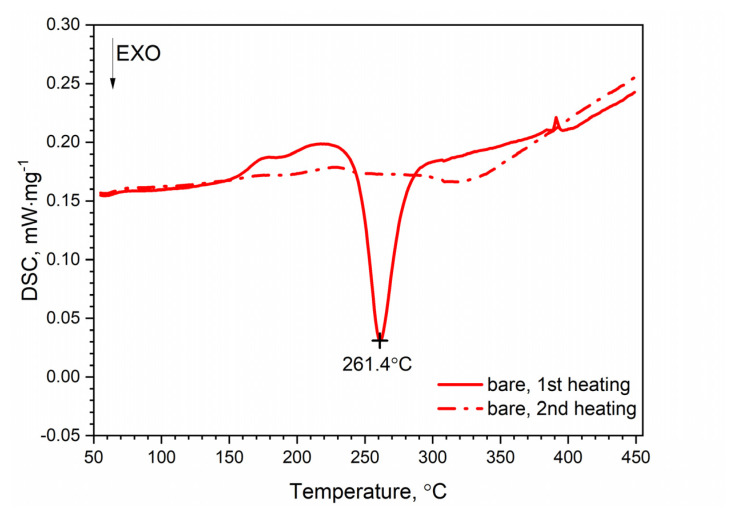
DSC thermograms of bare AW-2024-T3.

**Figure 8 materials-13-03345-f008:**
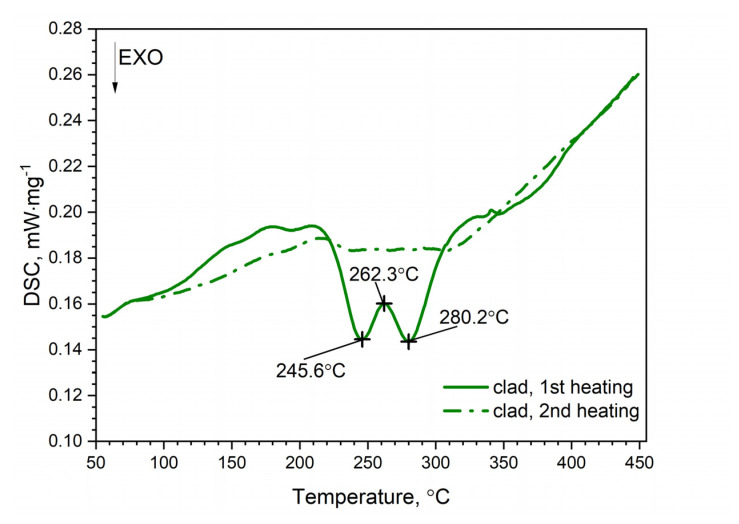
DSC thermograms of clad AW-2024-T3.

**Figure 9 materials-13-03345-f009:**
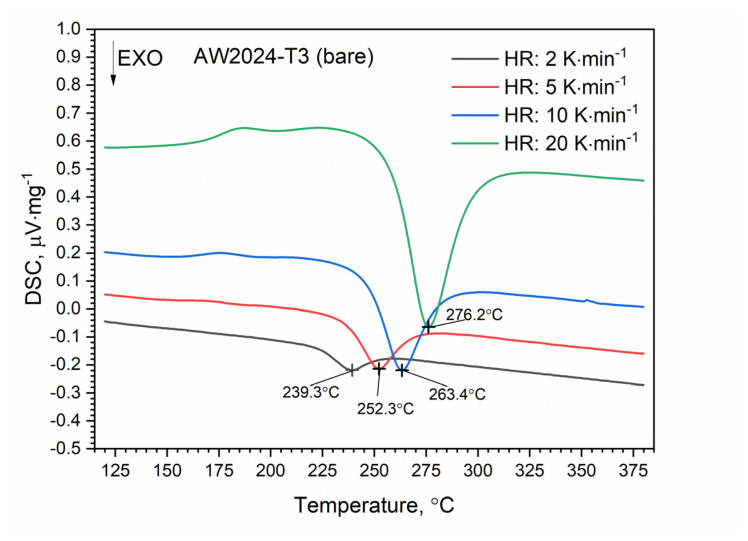
DSC thermograms of bare AW-2024-T3 at various heating rates (HRs).

**Figure 10 materials-13-03345-f010:**
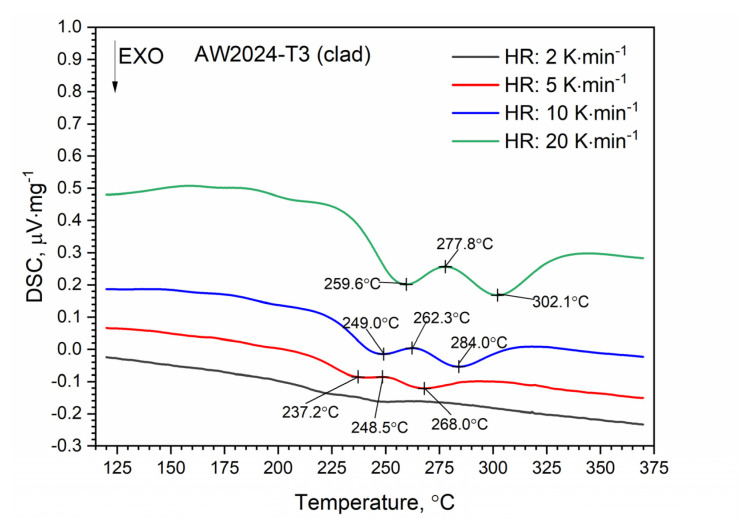
DSC thermograms of clad AW-2024-T3 at various HRs.

**Figure 11 materials-13-03345-f011:**
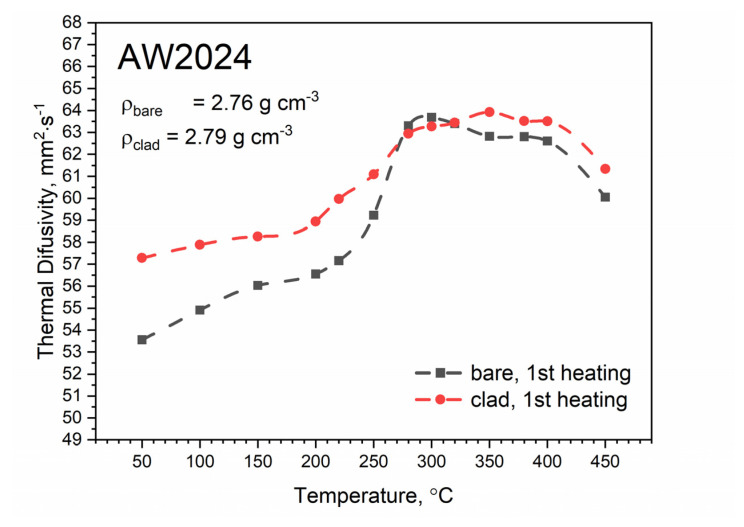
Thermal diffusivity as a function of temperature for bare and clad AW-2024-T3 alloys obtained from the first heating run on light flash apparatus (LFA) 467.

**Figure 12 materials-13-03345-f012:**
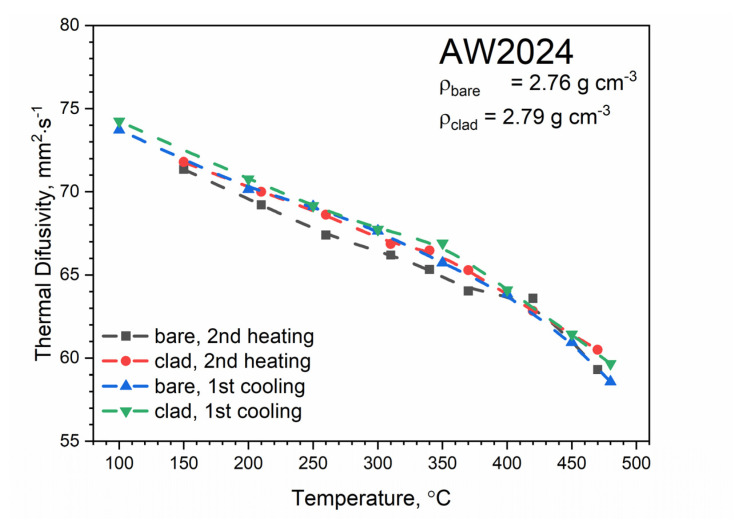
Thermal diffusivity as a function of temperature for bare and clad AW-2024-T3 alloys obtained from the first and second heating and first cooling runs on LFA 467.

**Figure 13 materials-13-03345-f013:**
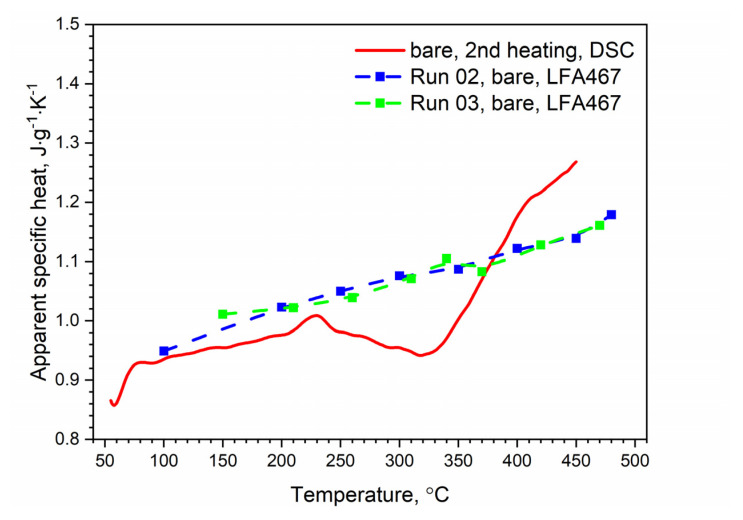
Comparison of the specific heat of the bare AW-2024-T3 alloy obtained from DSC (second heating run) and LFA measurements.

**Figure 14 materials-13-03345-f014:**
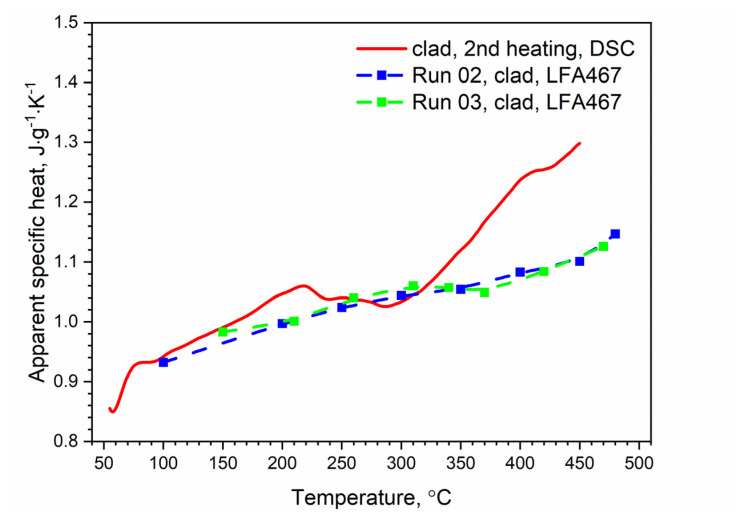
Comparison of the specific heat of the clad AW-2024-T3 alloy obtained from DSC (second heating run) and LFA measurements.

**Figure 15 materials-13-03345-f015:**
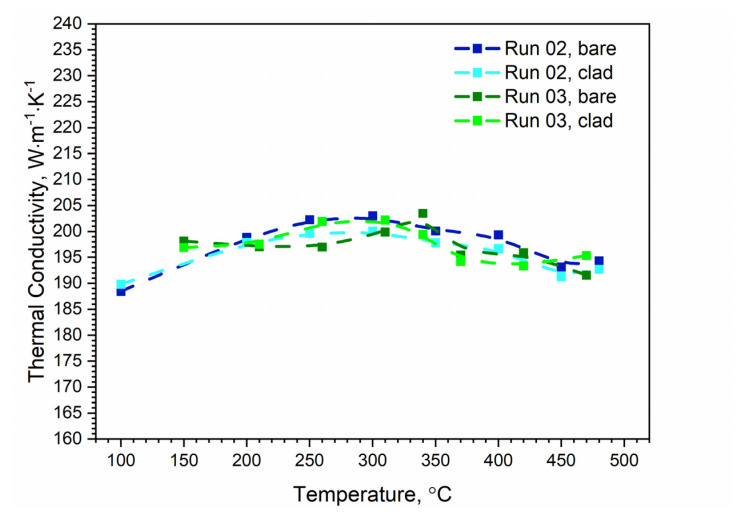
Comparison of the thermal conductivity of bare and clad AW-2024-T3 obtained from LFA measurements.

**Figure 16 materials-13-03345-f016:**
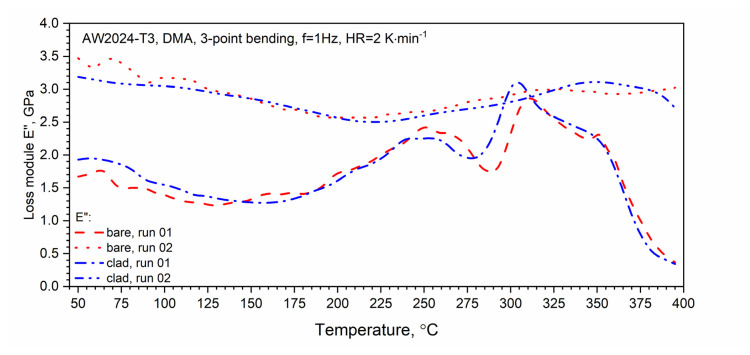
Dynamic loss modulus *E*″ of bare and clad AW-2024-T3 (first and second runs).

**Figure 17 materials-13-03345-f017:**
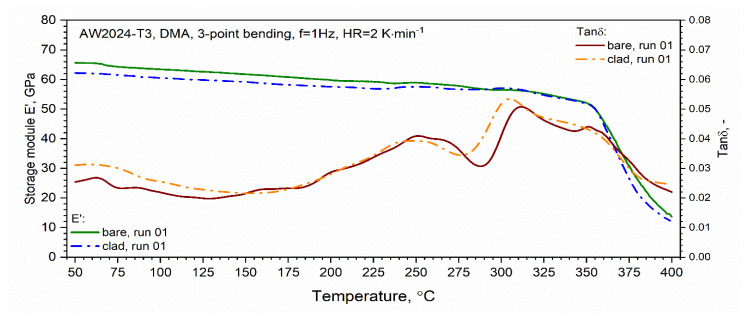
Dynamic storage modulus *E*′ and tan δ (dynamic dumper factor) of bare and clad AW-2024-T3 (first heating).

**Table 1 materials-13-03345-t001:** Chemical composition of the AW-2024 alloy [12].

Component	Si	Cr	Cu	Mg	Mn	Fe	Zn	Ti	Al
Concentration [wt.%]	0.00–0.50	0.00–0.10	3.80–4.90	1.20–1.80	0.3–0.90	0.00–0.50	0.00–0.25	0.00–0.20	bal.

**Table 2 materials-13-03345-t002:** Grain size and hardness of AW-2024-T3 alloy.

Sample	Microstructure/Global Shape	Grain size (Average ± Stdandard Deviation)		Ratio L/W	HV
		Length (µm)	Width (µm)		
AW-2024-T3 BARE					
Initial State	*Equiaxial*	~50	~50	1	118 ± 1
Heated	*Equiaxial*	~30	~30	1	70 ± 3
AW-2024-T3 CLAD					
Initial State	*Elongated*	~80	~40	2	125 ± 3
Heated	*Equiaxial*	-	-	-	59 ± 1

**Table 3 materials-13-03345-t003:** Chemical composition (in wt.% and at.%) of matrix and intermetallic precipitations in AW-2024-T3 alloy (The numbers of the precipitations in Table 3 refer to the numbers in Figure 2.

wt.% (At%)	Al	Cu	Mg	Mn	Si	Fe
AW-2024-T3 BARE INITIAL STATE						
*Matrix*	94.1 (95.6)	3.9 (2.3)	1.8 (2.0)	0.2 (0.2)	<0.1 (0.1)	-
*1*	57.7 (72.1)	11.4 (6.0)	-	11.4 (4.6)	5.1 (6.1)	18.3 (11.1)
*2*	91.7 (94.1)	4.35 (1.90)	1.49 (1.69)	0.3 (0.12)	2.3 (2.2)	-
*3*	7.4 (15.7)	92.1 (83.4)	0.19 (0.45)	0.2 (0.18)	-	0.2 (0.2)
*4*	36.2 (56.9)	63.1 (42.1)	0.51 (0.88)	<0.1 (0.1)	-	0.2 (0.1)
AW-2024-T3 HEATED						
*Matrix*	84.6 (92.6)	14.8 (6.9)	0.4 (0.4)	0.2 (0.1)	-	<0.1 (0.1)
*5*	75.6 (87.1)	22.6 (11.1)	1.1 (1.5)	0.5 (0.3)	0.2 (0.1)	-
*6*	14.6 (28.5)	84.3 (69.6)	0.2 (0.4)	-	0.7 (1.4)	0.1 (0.1)
*7*	66.7 (69.7)	5.5 (2.5)	0.8 (1.0)	<0.1 (0.1)	26.7 (26.7)	-
*8*	58.1 (76.1)	41.1 (22.8)	0.7 (1.1)	-	<0.1 (0.1)	-
AW-2024-T3 CLAD INITIAL STATE						
*Matrix*	95.5 (97.8)	3.9 (1.7)	0.4 (0.4)	0.3 (0.2)	-	-
*1*	58.4 (71.2)	7.0 (3.6)	-	9.9 (5.9)	8.0 (9.4)	16.7 (9.8)
*2*	38.0 (58.8)	61.2 (40.2)	0.3 (0.5)	<0.1 (0.1)	0.2 (0.3)	0.3 (0.2)
*3*	58.4 (71.1)	7.2 (3.7)	-	9.1 (5.5)	8.3 (9.7)	16.9 (9.9)
*4*	60.1 (72.2)	7.5 (3.8)	-	9.1 (5.4)	8.9 (10.2)	14.4 (8.3)
AW-2024-T3 CLAD HEATED						
*Matrix*	95.6 (97.9)	3.6 (1.56)	0.3 (0.3)	0.5 (0.2)	-	-
*5*	23.8 (42.3)	75.2 (56.6)	0.1 (0.3)	0.2 (0.2)	0.01 (0.2)	0.6 (0.5)
*6*	94.3 (97.3)	2.9 (1.3)	<0.1 (0.1)	1.1 (0.5)	-	1.6 (0.8)
*7*	86.5 (88.2)	2.1 (0.9)	0.1 (0.1)	0.3 (0.2)	10.9 (10.7)	-
*8*	94.3 (97.3)	2.9 (1.3)	0.1 (0.1)	1.1 (0.5)	-	1.6 (0.8)

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
