# Peer review of "Investigation of Thermophysical Properties of AW-2024-T3 Bare and Clad Aluminum Alloys"

_materials, 2020, doi:10.3390/ma13153345_

Round 1
Reviewer 1 Report
I read new version of manuscript and all my previous suggestions were incorporated. Therefore, I recommend the manuscript for publication.
Author Response
Paper Title:
Investigation of Thermophysical Properties of AW-2024-T3 Bare and Clad Aluminum Alloys
Authors: Janusz Zmywaczyk , Judyta Sienkiewicz, Piotr Koniorczyk, Jan Godzimirski, Mateusz Zieliński
Faculty of Mechatronics and Aerospace, Military University of Technology, ul. gen. S. Kaliskiego 2, 00-908
Warsaw, Poland
Re: Manuscript ID: materials-853368 (before: materials-838220)
Review 1
The authors would like to thank Reviewer 1 for his recommendation of our manuscript to be published in its current form.
Reviewer 2 Report
This submission cannot be accepted in the present form for several reasons:
- The authors state that there is an endothermic peak in Fig. 8. This is not the case, after the first exothermic peak, the curve just goes back up to the neutral line, but the second exo-peak sets in before the curve is all the way up.
- The authors have not stated that the numbers of the precipitations in table 3 refer to the numbers in Fig. 2, which I suppose is true.
- The authors introduce eq. (5) with "According to the most recent works", but do not include a citation.
- The units in Figs. 7 & 8 are different from those in Figs. 9 & 10; they should be the same.
- The English is such that some sentences cannot be understood, there are many mistakes in the English grammar; there are many punctuation mistakes; there are some typos and formatting mistakes.
Author Response
Paper Title:
Investigation of Thermophysical Properties of AW-2024-T3 Bare and Clad Aluminum Alloys
Authors: Janusz Zmywaczyk , Judyta Sienkiewicz, Piotr Koniorczyk, Jan Godzimirski, Mateusz Zieliński
Faculty of Mechatronics and Aerospace, Military University of Technology, ul. gen. S. Kaliskiego 2, 00-908
Warsaw, Poland
Re: Manuscript ID: materials-853368 (before: materials-838220)
Review 2
The authors would like to thank the Reviewer 2 for his valuable comments on our paper. We took time to reflect on the comments and changes suggested by the Reviewer 2. Changes have now been made in the light of these comments and modified text has been shown in blue colour in the amended paper.
|
-Reviewer 2 Actions taken |
|
|
1. The authors state that there is an endothermic peak in Fig. 8. This is not the case, after the first exothermic peak, the curve just goes back up to the neutral line, but the second exo-peak sets in before the curve is all the way up.
2. The authors have not stated that the numbers of the precipitations in table 3 refer to the numbers in Fig. 2, which I suppose is true.
3. The authors introduce eq. (5) with "According to the most recent works", but do not include a citation.
4. The units in Figs. 7 & 8 are different from those in Figs. 9 & 10; they should be the same.
5. The English is such that some sentences cannot be understood, there are many mistakes in the English grammar; there are many punctuation mistakes; there are some typos and formatting mistakes.
|
1. We thank You for this valuable comment. The following sentence has been included in the line 300: We observe two exothermic peaks in this temperature range, i.e. at 245.6℃ and 280.2℃ separated by a local peak at 262.3℃ which is not an endothermal peak but a relative maximum between two exothermal peaks.
2. It has been added in the text.
3. It has been corrected.
4. The unit mW/mg in Figures 7 and 8 results in the use of a sensitivity calibration curve to calculate the peak area in (J/g). Without calibration curve the DSC peak area is expressed in (mVs/mg). In case of Figures 9 & 10 unit (mV/mg) was used because the sensitivity calibration curves were not applied.
5. In case of obtaining a positive decision of our paper an extensive editing of English language and style will be directed to MDPI English Editing Section.
|
Reviewer 3 Report
The authors have greatly improved the previous submission by carefully following the suggestions of the reviewers.
Figures and Tables are correctly ordered and suitably cited in the manuscript.
The reviewer suggests to accept the manuscript after minor revisions, as reported in the following:
Lines 75-76: please, remove "(Electric Discharge Machining)". It is redundant.
Lines 397-398: please, remove the sentence "our test samples were commercially available".
Line 420: the use of references in the conclusions is not usual. The purpose of the conclusions is to highlight the critical points arised in the discussion. Therefore, comparison of the research findings with those in the literature should have been completed in the discussion. Please, remove.
Author Response
Paper Title:
Investigation of Thermophysical Properties of AW-2024-T3 Bare and Clad Aluminum Alloys
Authors: Janusz Zmywaczyk , Judyta Sienkiewicz, Piotr Koniorczyk, Jan Godzimirski, Mateusz Zieliński
Faculty of Mechatronics and Aerospace, Military University of Technology, ul. gen. S. Kaliskiego 2, 00-908
Warsaw, Poland
Re: Manuscript ID: materials-853368 (before: materials-838220)
Review 3
The authors would like to thank the Reviewer 3 for his valuable comments on our paper. We took time to reflect on the comments and changes suggested by the Reviewer 3. Changes have now been made in the light of these comments and modified text has been shown in green colour in the amended paper.
|
-Reviewer 3 Actions taken |
|
|
1. Lines 75-76: please, remove "(Electric Discharge Machining)". It is redundant. 2. Lines 397-398: please, remove the sentence "our test samples were commercially available". 3. Line 420: the use of references in the conclusions is not usual. The purpose of the conclusions is to highlight the critical points arised in the discussion. Therefore, comparison of the research findings with those in the literature should have been completed in the discussion. Please, remove. |
1. It has been done.
2. It has been done.
3. It has been done.
|
Reviewer 4 Report
Introduction: Generally well written, gives a good overview of previous work and states the objective of the presented research. Materials and Methods: Clearly written, with sufficient details. I observed some minor grammatical errors (e.g. L93: with an acceleration voltage at (at is not needed); or the degree signs in XRD measurement steps are typographically not correct) Results and Discussion: Section 3.1. describes the Al-Cu phase diagram. As it is written, this section clearly belongs rather to introduction than to results and discussion, as there is absolutely no work of the authors given in this section. Also, L176: According to the most recent works… and no reference ot these most recent works is given. L187-189 The α phase corresponds to the solid solution of Cu and the rest additives in the face centered cubic (FCC) lattice of Al while precipitates indicates the Al-Cu-X intermetallic (IM) compounds [22-27]. I don't understand this sentence. It would also be useful to label different microstructural components on the micrographs. L196: I think intermetallic is not a really long word and can be spelled out, instead of (another) abbreviation. Also, using IM for intermetallic is not really the norm. Figure 2 is very large, can it be broken maybe into two Figures, so that it does not take a whole page and a half? Also, it would be helpful to label the microstructure components and the LM micrographs are not really good (Overetched samples? Bad whitebalance? Underfocused?– do you have any better images???) Diffractogram in Figure 5: even though different symbols were used for the peak identification, it is hard to see which is which. Also, since the Al peak is so large and the other peaks belonging to the phases are smaller, it would be better to cut out the most intense peak, so the smaller peaks could be seen. Since the results of this measurements are only briefly mentioned in passing, I am wondering if it is really necessary or can it be omitted – similar fort he EDS mappings. The article is already quite long and microstructure was not really the focal point of the manuscript. Table 3: Since the reference to table 3 is given quite a few pages away, it would be useful to say that this is the result of EDS analysis. Also, unless you have a very well calibrated EDS system and used standards of similar composition, the two decimal places for EDS analysis is an overkill. One decimal space is enough. The description of microstructures is quite long and tedious. I suggest some editing for style and clarity of the presentation of results. Section 3.4.1. needs some editing. There are some grammar mistakes, typographical mistakes (like the placement of degree signs for temperatures) and the style of the section is also quite poor. For example, i.e. is used much too often and unnecessarily. Conclusions: L394-398 make no sense and are not conclusions. Again, some editing for style would be needed. References: Unfortunately, I could not access ref. 37 (Military handbook) to compare the diagrams of thermal diffusivity and specific heats. Also, reference 17 (phase diagram) points to a Wikimedia file - which is also used in the manuscript. The editors should decide if this is OK.
Author Response
Paper Title:
Investigation of Thermophysical Properties of AW-2024-T3 Bare and Clad Aluminum Alloys
Authors: Janusz Zmywaczyk , Judyta Sienkiewicz, Piotr Koniorczyk, Jan Godzimirski, Mateusz Zieliński
Faculty of Mechatronics and Aerospace, Military University of Technology, ul. gen. S. Kaliskiego 2, 00-908
Warsaw, Poland
Re: Manuscript ID: materials-853368 (before: materials-838220)
Review 4
The authors would like to thank the Reviewer 3 for his valuable comments on our paper. We took time to reflect on the comments and changes suggested by the Reviewer 3. Changes have now been made in the light of these comments and modified text has been shown in violet colour in the amended paper.
|
-Reviewer 4 Actions taken |
|
|
1. Materials and Methods: Clearly written, with sufficient details. I observed some minor grammatical errors (e.g. L93: with an acceleration voltage at (at is not needed); or the degree signs in XRD measurement steps are typographically not correct. 2. Results and Discussion: Section 3.1. describes the Al-Cu phase diagram. As it is written, this section clearly belongs rather to introduction than to results and discussion, as there is absolutely no work of the authors given in this section. 3. Also, L176: According to the most recent works… and no reference of these most recent works is given
4. L187-189 The α phase corresponds to the solid solution of Cu and the rest additives in the face centered cubic (FCC) lattice of Al while precipitates indicates the Al-Cu-X intermetallic (IM) compounds [22-27]. I don't understand this sentence. It would also be useful to label different microstructural components on the micrographs. 5. L196: I think intermetallic is not a really long word and can be spelled out, instead of (another) abbreviation. Also, using IM for intermetallic is not really the norm. Figure 2 is very large, can it be broken maybe into two Figures, so that it does not take a whole page and a half? Also, it would be helpful to label the microstructure components and the LM micrographs are not really good (Overetched samples? Bad whitebalance? Underfocused?– do you have any better images???). 6. Diffractogram in Figure 5: even though different symbols were used for the peak identification, it is hard to see which is which. Also, since the Al peak is so large and the other peaks belonging to the phases are smaller, it would be better to cut out the most intense peak, so the smaller peaks could be seen. Since the results of this measurements are only briefly mentioned in passing, I am wondering if it is really necessary or can it be omitted – similar fort he EDS mappings. 7. The article is already quite long and microstructure was not really the focal point of the manuscript. Table 3: Since the reference to table 3 is given quite a few pages away, it would be useful to say that this is the result of EDS analysis. Also, unless you have a very well calibrated EDS system and used standards of similar composition, the two decimal places for EDS analysis is an overkill. One decimal space is enough. The description of microstructures is quite long and tedious. I suggest some editing for style and clarity of the presentation of results. 8. Section 3.4.1. needs some editing. There are some grammar mistakes, typographical mistakes (like the placement of degree signs for temperatures) and the style of the section is also quite poor. For example, i.e. is used much too often and unnecessarily.
9. Conclusions: L394-398 make no sense and are not conclusions. Again, some editing for style would be needed. 10. References: Unfortunately, I could not access ref. 37 (Military handbook) to compare the diagrams of thermal diffusivity and specific heats. Also, reference 17 (phase diagram) points to a Wikimedia file - which is also used in the manuscript. The editors should decide if this is OK.
|
1. It has been corrected.
2. We suggest leaving this Al-Cu phase diagram in Chapter 3.1 so as not to completely change the order of our manuscript. The authors believe that Figure 1 is a natural introduction to the subject matter discussed in this chapter.
3. This sentence has been rewritten as: According to the most recent work these sequence can be written as: αsss → GPB zones → S” → S’ → S-Al2CuMg (5) with GPB being Guinier-Preston-Bagaryatsky zones (Cu-Mg co-clusters), S” and S’ being metastable phases of Al2CuMg and S-Al2CuMg being equilibrium (stable) precipitates [13].
4. The sentence has been changed. The α phase corresponds to the solid solution of Cu and the rest additives in the face centered cubic (FCC) lattice of Al while observed precipitates indicates the Al-Cu-X intermetallic compounds [22-27]. Microstructural components has been labeled.
5. It has been corrected. Some Figures has been omitted. Microstructure components has been labeled.
6. We suggest leaving this XRD pattern and EDS mappings so as not to completely change the order of our manuscript. The XRD pattern has been corrected.
7. To shorten the article some images has been deleted. Table 3 has been corrected.
8. The text has been changed as follows: Lines 300-302: We observe two exothermic peaks in this temperature range, i.e. at 245.6℃ and 280.2℃ separated by a local peak at 262.3℃ which is not an endothermal peak but a relative maximum between two exothermal peaks. Lines 306-309: Each time, i.e. For each HR a new sample was taken for testing. The apparent activation energy of q’ - phase corresponds to 137.1 ± 4.4 kJ mol-1 for bare and to 131.0 ± 6.0 (exo), 104.1 ± 2.1 (exo), (two peaks – Figure 10) for clad alloy.
9. These sentences were added to an earlier version of the text in response to the comments of previous Reviewers (1-3).
10. Ref. 37 can easily be obtained in the Knovel database (see - screenshots - access dated June 25, 2020). Copyright permission for reference 17 is not required according to our knowledge. |
Round 2
Reviewer 2 Report
This submission cannot be accepted for publication, mainly because the presentation is really lousy. Apart from that, there is hardly any discussion - this would have been better with an own discussion section.
Author Response
|
1. This submission cannot be accepted for publication, mainly because the presentation is really lousy. Apart from that, there is hardly any discussion - this would have been better with an own discussion section
|
1. We appreciate Your comment concerning discussion. In the chapter 4 - Conclusion the following text has been added:
The results of microstructures and composition investigations of AW-2024-T3 bare and clad aluminum alloys are summarized as follows: - at initial state Al2Cu and Al2CuMg intermetallics enriched in Mn, Fe and Si are confirmed in the both alloys; - after DSC measurements precipitates were coarser and were present inside grains as well as on grain boundaries; - a significant decrease in hardness was observed for both alloys after DSC tests. The results of the kinetics investigations of the both alloys recorded during heating confirm the precipitation/dissolution sequences within the following temperature ranges: - DSC thermograms: a) 150 – 245 ℃ (bare) and 100 – 225 ℃ (clad) GP and GPB zones dissolve; b) 245 – 285 ℃ (bare) and 225 – 305 ℃ (clad) θ’ and S’ phase precipitate; c) 285 – 390 ℃ (bare) and 305 – 340 ℃ (clad) θ' and S' phase dissolve; d) above 390 ℃ (bare) and 340 ℃ (clad) the stable equilibrium θ-Al2Cu and S-Al2CuMg phase forms. - DMA thermograms: a) 150 – 250 ℃ (bare and clad) GP and GPB zones dissolve; b) 250 – 290 ℃ (bare) and 250 – 280 ℃ (clad) θ’ and S’ phase precipitate; c) 290 – 310 ℃ (bare) and 290 – 305 ℃ (clad) θ' and S' phase dissolve; d) 310 – 350 ℃ (bare) and 305 – 350 ℃ (clad) fine phases (~10 nm) q and S precipitate; e) above 350 ℃ (bare and clad) the stable equilibrium θ-Al2Cu and S-Al2CuMg phase forms and rapid decrease of bare and clad alloys stiffness (E’) is observed. The results of the thermal properties studies are summarized as follows: - LFA thermograms for the first heating: a) 50 – 300 ℃ (bare and clad) – monotonic increase of thermal diffusivity from 53 mm2×s-1 (bare) and 57 mm2×s-1 (clad) up to 64 mm2×s-1 (for both alloys); b) 300 – 450 ℃ (for both alloys) monotonic decrease of thermal diffusivity down to 60 mm2×s-1; - LFA thermograms for the second and subsequent heating: a) 100 – 430 ℃ (for both alloys) monotonic decrease of thermal diffusivity from 74 mm2×s-1 down to 58 mm2×s-1; - specific heat determined from LFA investigations (second and subsequent heating) using comparative method and from DSC studies: An increase in the specific heat value was observed, within the temperature range 100 – 480 ℃ comparable for both alloys. Slightly discrepancies between DSC and LFA specific heat results did not exceed 10% were observed; - thermal conductivity from LFA investigations (second and subsequent heating) using comparative method: In the temperature range 100 – 480 ℃ (for both alloys) thermal conductivity slightly changes from 190 to 200 W×m-1×K-1. |
Reviewer 4 Report
The authors have improved their manuscript according to suggestions and I have no further comments at this point.
Author Response
The authors would like to thank Reviewer 4 for his recommendation of our manuscript to be published in its current form.